# Revisiting the question “Why is the sky blue?”

Anna Lange [1], Alexei Rozanov [2], and Christian von Savigny [1]

[1]Institute of Physics, University of Greifswald, Felix-Hausdorff-Str. 6, 17489 Greifswald, Germany

[2]Institute of Environmental Physics, University of Bremen, Otto-Hahn-Allee 1, 27359 Bremen, Germany

**Correspondence:** Anna Lange (anna.lange@uni-greifswald.de)

**Abstract.** The common answer to the question "Why is the sky blue" is usually Rayleigh scattering. In 1953 Edward Hulburt demonstrated, that the blue colour of the zenith sky at sunset is to 1/3 caused by Rayleigh scattering and to 2/3 caused by ozone absorption. In this study, an approach to quantify the contribution of ozone to the blue colour of the sky for different viewing geometries is implemented using the radiative transfer model SCIATRAN and the CIE (International Commission on Illumination) XYZ 1931 colour system. The influence of ozone on the blue colour of the sky is calculated for solar zenith angles of 10° – 90° and a wide range of viewing geometries. For small solar zenith angles, the influence of ozone on the blue colour of the sky is minor, as expected. However, the effect of ozone increases with increasing solar zenith angle. The calculations for the Sun at the horizon confirm Hulburt’s estimation with remarkably good agreement. More aerosols reduce the ozone contribution at and near the zenith for the Sun at the horizon. The exact contribution of ozone depends strongly on the assumed total ozone column. The calculations also show that the contribution of ozone increases with increasing viewing zenith angle and total ozone column. Variations in surface albedo as well as full treatment of polarised radiative transfer were found to have only minor effects on the contribution of ozone to the blue colour of the sky. Furthermore, with an observer at 10 km altitude an increase of the ozone influence can be seen.

## 1 Introduction

The blue colour of the cloud-free sky is one of the most obvious and self-evident features of our natural environment and has puzzled humans for thousands of years. The blue colour of the sky is nowadays usually explained by Rayleigh scattering, an explanation that is, however, not entirely correct. In his 1953 publication entitled “Explanation of the Brightness and Color of the Sky, Particularly the Twilight Sky”, Edward Olson Hulburt (1890 – 1982) demonstrated that for specific illumination and viewing conditions, Rayleigh scattering plays only a second order role for the blue colour of the sky. Based on simplified radiative transfer simulations and single scattering approximation, Hulburt (1953) concluded that for a solar zenith angle (SZA) of 90° the blue sky of the zenith is caused to only $1/3$ by Rayleigh scattering and to $2/3$ by absorption of solar radiation in the Chappuis bands of $O_3$. Interestingly, Hulburt (1953) did not explain, how these contributions of Rayleigh scattering and $O_3$ absorption were determined. However, it can be assumed that the estimation is based on the spectra calculated by Hulburt (Hulburt, 1953, Fig. 5). In the present paper we define a metric that allows determining the effect of $O_3$ absorption on the blue sky colour in a quantitative way.

In the following we briefly review different explanations for the blue colour of the sky that were suggested in the past. According to See (1904), Leonardo da Vinci suggested that the blue colour of the sky is caused by mixing sunlight with the black colour of space. Newton was of the opinion that the blue of the sky was caused by small water particles that "reflect" blue light (Newton, 1730). Their size was assumed to be so small as to leave the atmosphere transparent. Newton expected larger particles to reflect different colours. Newton's theory was generally accepted well into the 19th century. An important realisation of the first half of the 19th century was that of the polarisation of sunlight scattered in the atmosphere by Francois Arago in 1809 (Hoeppe, 2007). Of essential importance for the understanding of the blue sky colour were the laboratory experiments by John Tyndall, e.g. described in Tyndall (1869). Tyndall produced tiny particles of many different compositions in a glass cylinder and observed the colour and the polarisation of scattered radiation. He found: "In all cases, and with all substances, the cloud formed at the commencement, when the precipitated particles are sufficiently fine, is blue [...]." (Tyndall, 1869). Furthermore, he found that these blue clouds in all cases completely polarised the scattered radiation with the direction of polarisation and the propagation direction of the incident beam forming a right angle. In 1871 John William Strutt (Baron Rayleigh III from 1873) provided a theory for both the polarisation and the spectral dependence of radiation scattered by particles which are small compared to the wavelength (Strutt, 1871). Regarding the spectral dependence, he first gave a derivation of the famous $\lambda^{-4}$-law solely based on dimensional analysis, followed by an analytical derivation. Strutt (1871) also provided a comparison of his theoretical considerations with observations of the spectral dependence of the intensity of zenith scattered sunlight, showing good agreement. However, the nature of the small particles causing the blue colour of the sky and its polarisation was not yet established. Only in 1899 Lord Rayleigh III demonstrated that scattering by air molecules is sufficient to explain the observed effects (Rayleigh, 1899).

In 1974 Adams et al. investigated the influence of ozone and aerosols on the brightness and colour of the twilight sky, using five different models of the atmosphere, such as a pure molecular scattering atmosphere, a molecular atmosphere with ozone and three models with varying aerosol concentrations. For this purpose, the authors calculated the radiance and colour of the twilight sky for solar zenith angles of $90° – 96°$ assuming single scattering (more information in Sec. 3). Adams et al. (1974), as well as the following papers: e.g. Gadsden (1957), Dave and Mateer (1968), and Lee et al. (2011) focused on twilight conditions, i.e. solar zenith angles greater than 90°, and the zenith sky observations. The investigations in the present paper cover solar zenith angles of $10° – 90°$ with a wide range of viewing geometries.

The paper is structured as follows. In Sec. 2 the radiative transfer model SCIATRAN, the colour representation in the CIE chromaticity diagram and the method for determining the ozone contribution to the blue colour of the sky are introduced. Section 3 presents the main results with discussion and the conclusions are given at the end (Sec. 4).

## 2 Methodology

### 2.1 Radiative transfer simulations with SCIATRAN

For the radiative transfer simulations, the SCIATRAN software package (version 4.5.5) with implemented Mie code was used (Rozanov et al., 2014). The simulations were performed in the approximate spherical mode ("approximate method" in the

following). In this mode, the contribution of single scattering is calculated in a fully spherical geometry, whereas an approximation is used to account for the multiple scattering contribution (Rozanov et al., 2000). The multiple scattering radiative field was initialised by using the discrete ordinate technique. Figure 1 shows the ratio of scattered solar radiance spectra for the exact method (fully spherical mode) and the approximate method for SZA (solar zenith angle) = 90°, SAA (solar azimuth angle) = 0° (corresponding to the sun-ward direction) and different VZAs (viewing zenith angles) for an observer on the Earth's surface. The differences are primarily in the (short-wave) blue spectral range. Calculations were also performed for VZA = 90° – with the resulting ratios covering a range from ≈ 0.997 - 1.014 (not shown). Overall, the approximate method is completely sufficient for the simulations carried out here.

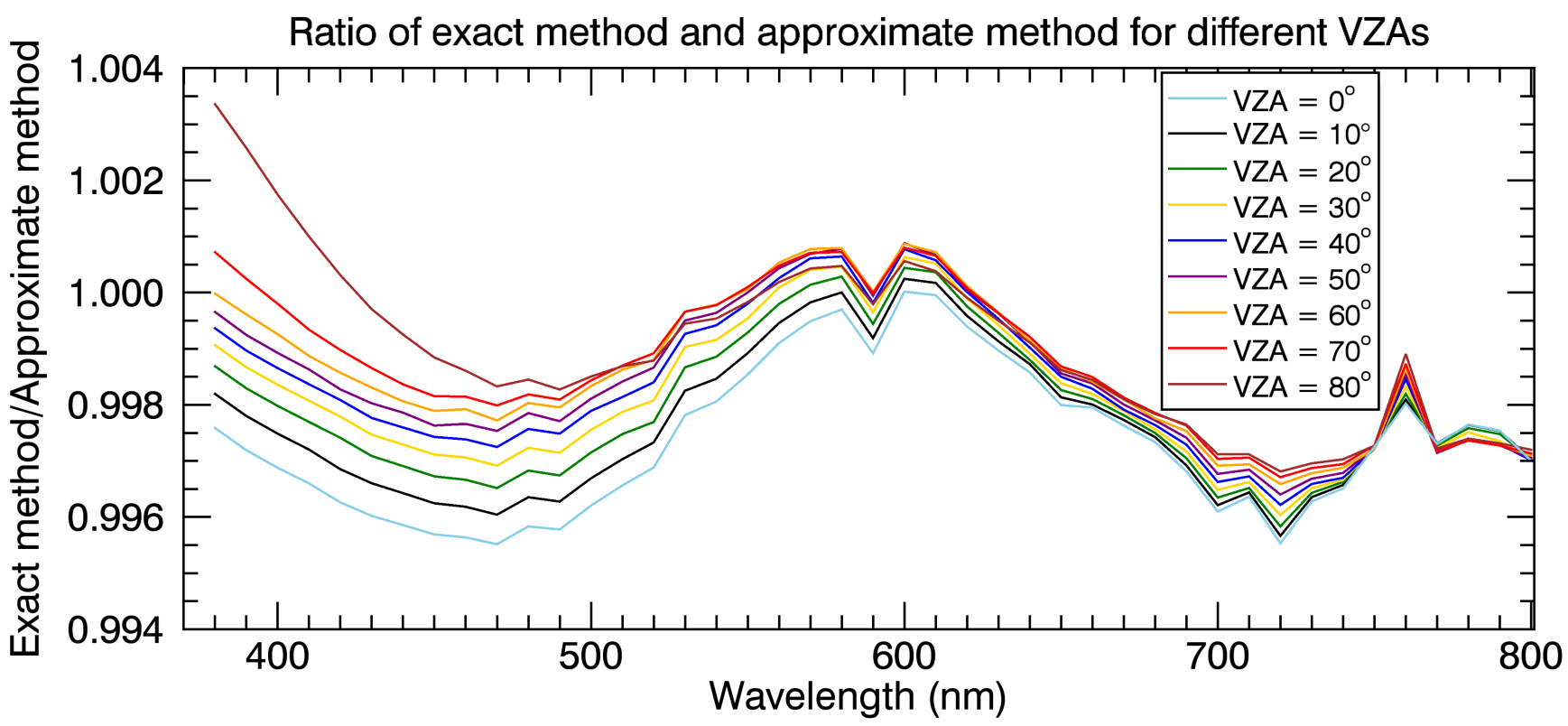

**Figure 1.** Ratio of scattered solar radiance spectra for the exact and approximate method for different VZAs, SZA = 90° and SAA = 0°.

For the simulations with SCIATRAN, vertical profiles of atmospheric trace gases, pressure and temperature for northern mid-latitudes were used from the incorporated climatological database based on a 3-D chemical transport model (Sinnhuber et al., 2003). The effects of the refraction are considered for both direct and scattered sunlight. The calculations were done neglecting the polarisation. A test considering polarisation for SZAs = 10° and 70° and different viewing geometries resulted in a maximum relative difference of ≈ 1 % of the $x$, $y$ chromaticity coordinates.

Simulations were performed for the following viewing geometries: SZAs = 10°, 30°, 50°, 70°, 80°, 90°, VZAs = 0° – 90°, in 10° steps and SAAs = 0° – 180°, in 30° steps.

A mono-modal log-normal particle size distribution was assumed for the implemented tropospheric and stratospheric aerosols:

$$n(r) = \frac{N_0}{\sqrt{2\pi} \cdot \ln(S) \cdot r} \cdot \exp\left[-\frac{(\ln r - \ln r_m)^2}{2\ln^2(S)}\right], \tag{1}$$

where $N_0$ is the total particle number density, $S$ the geometric standard deviation of the distribution, $r$ the particle radius and $r_m$ the median radius. The corresponding input parameters for the Mie calculations carried out with the Mie code included in

SCIATRAN are discussed in the following Sec. 3. Output of the simulations performed here are the intensities of the radiation in a wavelength range of 380 - 800 nm. In order to obtain the spectral distribution of the scattered solar radiation for an observer on the Earth's surface, the simulated spectra were multiplied with a solar irradiation spectrum (SORCE (Solar Radiation and Climate Experiment) data (LASP, 2003)). For more detailed information on the radiative transfer model SCIATRAN we refer to, e.g., Rozanov et al. (2014).

### 2.2 Chromaticity coordinates

The CIE (International Commission on Illumination) XYZ 1931 colour system enables the objective representation of colours in a 2-dimensional coordinate system with the chromaticity coordinates $x$ and $y$ on the axes, the so-called CIE chromaticity diagram (e.g., Wyszecki and Stiles, 2000; CIE, 2004). Using the CIE colour matching functions $\overline{x}(\lambda)$, $\overline{y}(\lambda)$ and $\overline{z}(\lambda)$, which describe the spectral sensitivity of the cone cells of the human eye, the CIE XYZ tristimulus values of a simulated spectrum $I(\lambda)$ were determined by multiplying the simulated spectrum by these functions with subsequent integration (Billmeyer Jr. and Fairman, 1987):

$$X = \int_{380\,\mathrm{nm}}^{800\,\mathrm{nm}} I(\lambda)\overline{x}(\lambda)\,d\lambda \tag{2}$$

$$Y = \int_{380\,\mathrm{nm}}^{800\,\mathrm{nm}} I(\lambda)\overline{y}(\lambda)\,d\lambda \tag{3}$$

$$Z = \int_{380\,\mathrm{nm}}^{800\,\mathrm{nm}} I(\lambda)\overline{z}(\lambda)\,d\lambda \tag{4}$$

The chromaticity values $x$ and $y$ were then calculated as follows:

$$x = \frac{X}{X+Y+Z} \qquad y = \frac{Y}{X+Y+Z} \tag{5}$$

For the representation in the CIE chromaticity diagram, the CIE XYZ tristimulus values were converted to standard RGB (sRGB). However, it should be noted that the displayed colours may vary depending on the output device. More information on colour modelling can be found in previous papers, e.g., Lange et al. (2023) and Wullenweber et al. (2021).

### 2.3 Determining the contribution of ozone to the blue colour of the sky

To determine the contribution of ozone to the blue colour of the sky for different viewing geometries, the corresponding chromaticity coordinates $x$ and $y$ were first calculated. Figure 2 shows an example of the representation in the CIE chromaticity diagram (enlarged) with SZA = 10°, SAA = 0°, VZAs = 10°, 80°, 86°, 90° (left panel) and SZA = 90°, SAA = 0°, VZAs = 10°, 40°, 60°, 70° (right panel) - both for an observer on the Earth's surface. The "x" in the CIE chromaticity diagram ($x$ = $y$ = 1/3) corresponds to the "white point".

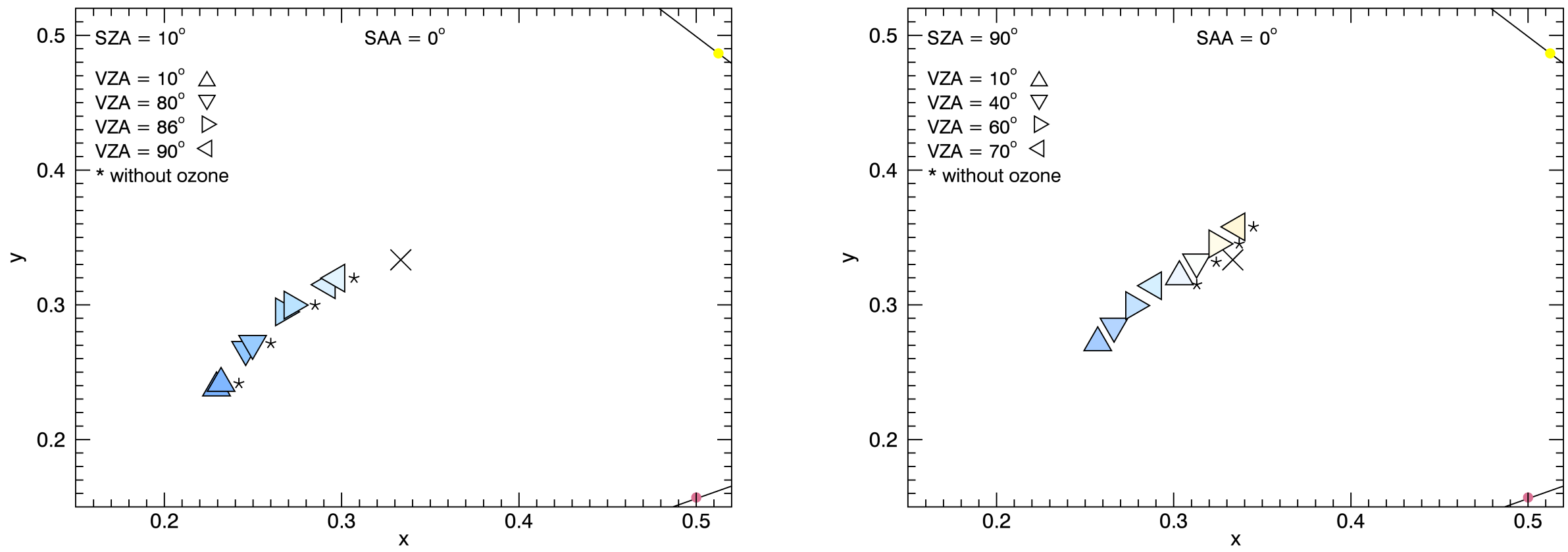

**Figure 2.** Example of the representation in the CIE chromaticity diagram (enlarged). Left panel: SZA = 10°, SAA = 0°, VZAs = 10°, 80°, 86°, 90°. Right panel: SZA = 90°, SAA = 0°, VZAs = 10°, 40°, 60°, 70°. Both for an observer on the Earth's surface.

To estimate the contribution of ozone to the blue colour of the sky, the next step was to calculate the distances between the data point (with ozone $d_1$ and without ozone $d_2$) and the "white point" in the CIE chromaticity diagram. Subsequently, the difference of both distances was determined, divided by the distance $d_1$ (with ozone) and multiplied by 100 % ($d_1$ is always greater than $d_2$):

$$\frac{\mathrm{d_1 - d_2}}{\mathrm{d_1}} \cdot 100\,\% = \text{relative difference} \tag{6}$$

The calculated relative differences are colour-coded and displayed in polar diagrams in the following sections.

The method is not applicable for all cases. The right panel of Fig. 2 shows for e.g. VZAs = 60° and 70° viewing geometries where the method is inapplicable. That is generally the case for chromaticity coordinates corresponding to the green, yellow, orange and red colour regions of the CIE chromaticity diagram and can apply to one data point or both (with and without ozone) – this covers all cases occurring in this work but not every theoretically possible one. Since the studies focus on the blue colour of the sky, these limitations occur mainly at large SZAs and at VZAs near the horizon. Deviations from this can appear due to changes in the total ozone column (TOC) and aerosol content (further discussed in the following section). Note that the fact that the method is not applicable in all cases does not affect the goals of the study, i.e. to determine the contribution of ozone to the blue colour of the sky, because the sky is not blue anymore in the cases that cannot be studied.

## 3 Results and discussion

The results in this section are based on the following assumptions, if not stated otherwise. For the particle size distribution and spectral extinction properties of the implemented tropospheric aerosol layer in the altitude range of 0 - 9 km, the "tropospheric" aerosol model from Shettle and Fenn (1979) was used. The aerosols consist of 70 % water-soluble components (ammonium and calcium sulphate and organics) and 30 % dust-like components with the following parameters: $r_m$ = 33 nm, $S$ = 1.4 and an aerosol optical depth (AOD) at 550 nm of 0.04 (Shettle and Fenn, 1979). The AOD of 0.04 at 550 nm is in the range of AOD values observed with the AERONET (Aerosol Robotic Network) photometer at the Institute of Physics of the University of Greifswald. In Shettle and Fenn (1979), the AOD is not predefined and can be adjusted accordingly using the number density $N$, which is normalised to 1 particle/cm$^3$. The mono-modal log-normal particle size distribution of the tropospheric aerosols results from the elimination of larger particles from the model, since the particles above the boundary layer have a longer residence time and the larger particles settle due to gravity (Shettle and Fenn, 1979). For clear conditions, the "tropospheric model" is also valid for the boundary layer. More detailed information can be found in Shettle and Fenn (1979). It should be noted that the aerosol content in the troposphere is highly variable, and it is not possible to account for this variability within the scope of this work.

The stratospheric aerosol layer in 10 - 25 km altitude consists of sulphate particles with the following characteristics: $r_m$ = 100 nm, $S$ = 1.6 and AOD (550 nm) = 1.4 $\cdot 10^{-3}$. Other AODs for the stratosphere are also used in the following – see more details below. Accordingly, the total AOD of both aerosol layers is 0.0414 at 550 nm. For the TOC, 300 DU was first assumed.

Figure 3 shows polar diagrams for SZAs = 10° (a), 30° (b), 50° (c), 70° (d), 80° (e) and 90° (f) with the relative differences (see equation 6) colour-coded. The radius of the polar diagram corresponds to the VZA and the angle to the SAA (marked at the outer edge of the diagram). The upside down triangle symbol in the diagrams indicates the observation geometry where the ground-based observer is looking directly into the Sun. Note that the simulations performed here consider only the scattered and not the directly transmitted sunlight. Furthermore, due to the azimuthal symmetry of the radiation field, the SAAs between 180° and 360° lead to the same simulation results as the corresponding SAAs between 180° and 0°. Therefore, only results for SAAs between 0° and 180° are shown in the following. Missing values within this SAA range are due to the non-applicability of the method.

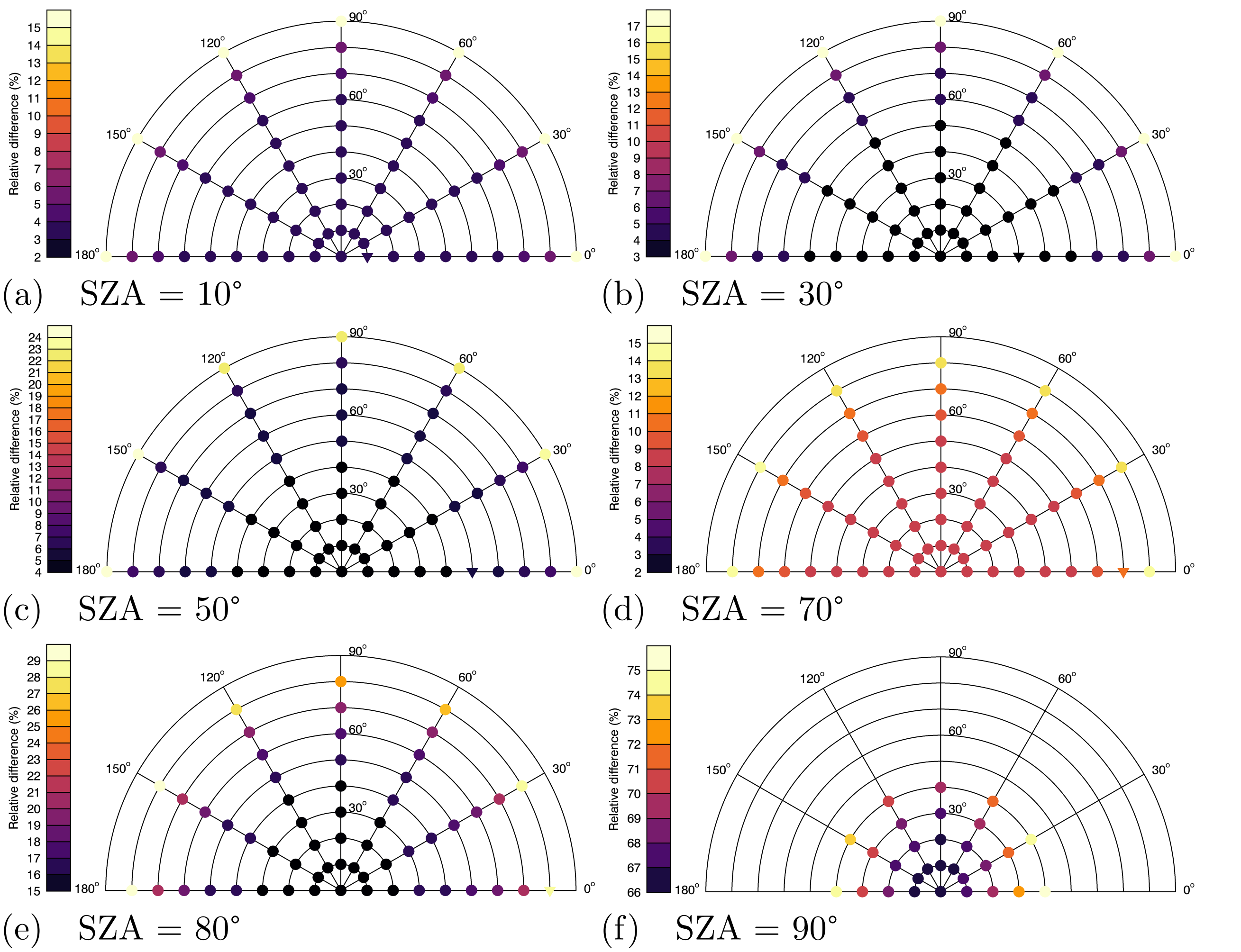

**Figure 3.** Polar diagrams for SZAs = 10° (a), 30° (b), 50° (c), 70° (d), 80° (e) and 90° (f) with the relative differences colour-coded. The missing values are due to the non-applicability of the method as described above.

For SZA = 10° (panel (a) of Fig. 3) small relative differences are found as expected, i.e. a minor influence of ozone on the blue colour of the sky, which increases with increasing VZA from 3 % (VZA = 0°) to 15 % (VZA = 90°). The first change of the relative differences can be seen at VZA = 70°. Similar results are observed for SZA = 30° (panel (b) of Fig. 3) with 3 % relative difference up to VZA = 50° and 17 % at VZA = 90°. For both SZAs, no SAA-dependent change of the ozone influence is found.

At larger SZAs, i.e. longer light path through the atmosphere, the ozone influence increases and thus the values of the relative difference (compare panels (c) – (f) of Fig. 3). For SZA = 50°, the contribution of ozone is 4 % at the zenith (VZA = 0°) (up

to VZA = 40°) and 24 % at the horizon (VZA = 90°) (panel (c) of Fig. 3). The relative difference for VZA = 90° varies here slightly with the SAA, for SAAs = 0°, 150°, 180° the value is 24 %, for SAA = 30° it is 23 % and for SAAs = 60°, 90° and 120° the value of the relative difference is 22 %. In comparison, the contribution of ozone to the blue colour for a SZA of 70°, is 8 % at the zenith (up to VZA = 50°) and ≈ 14 % at VZA = 80° (for SZA = 50°, VZA = 80° it is 8 %). The value of the relative difference at VZA = 80° also varies with the SAA here (see panel (d) of Fig. 3).

The panels (e) and (f) of Fig. 3 show polar diagrams for SZA = 80° (e) and SZA = 90° (f). The values of the relative difference calculated for SZA = 80° range from 15 % (VZA = 0°) to ≈ 29 % (VZA = 80°) – depending on the SAA. For SZA = 90° the relative differences and thus the contribution of ozone increase significantly. At the zenith, ozone contributes 66 % to the blue colour. This is in good agreement with Hulburt's 1953 estimate of 2/3 ozone influence at the zenith at sunset with a TOC of 240 DU. Our calculations for TOC = 240 DU, SZA = 90°, VZA = 0° and SAA = 0° yielded a value of 60 % (not shown). However, this is still close to the ozone contribution determined by Hulburt. Also for SZA = 90°, the contribution of ozone increases with increasing VZA with a maximum value of 75 % at VZA = 40° and SAA = 0°. The relative differences for the SAAs of 30° - 150° are slightly smaller compared to SAA = 0° and 180°. Figure 4 shows the corresponding scattered solar radiation spectra for SZA = 90° and VZA = 0° with TOC = 300 DU (black line) and TOC = 0 DU (green line). The effect of the ozone Chappuis bands (centred around 600 nm) on the scattered radiance spectra is quite pronounced, consistent with the large influence of ozone on the sky colour of this viewing geometry.

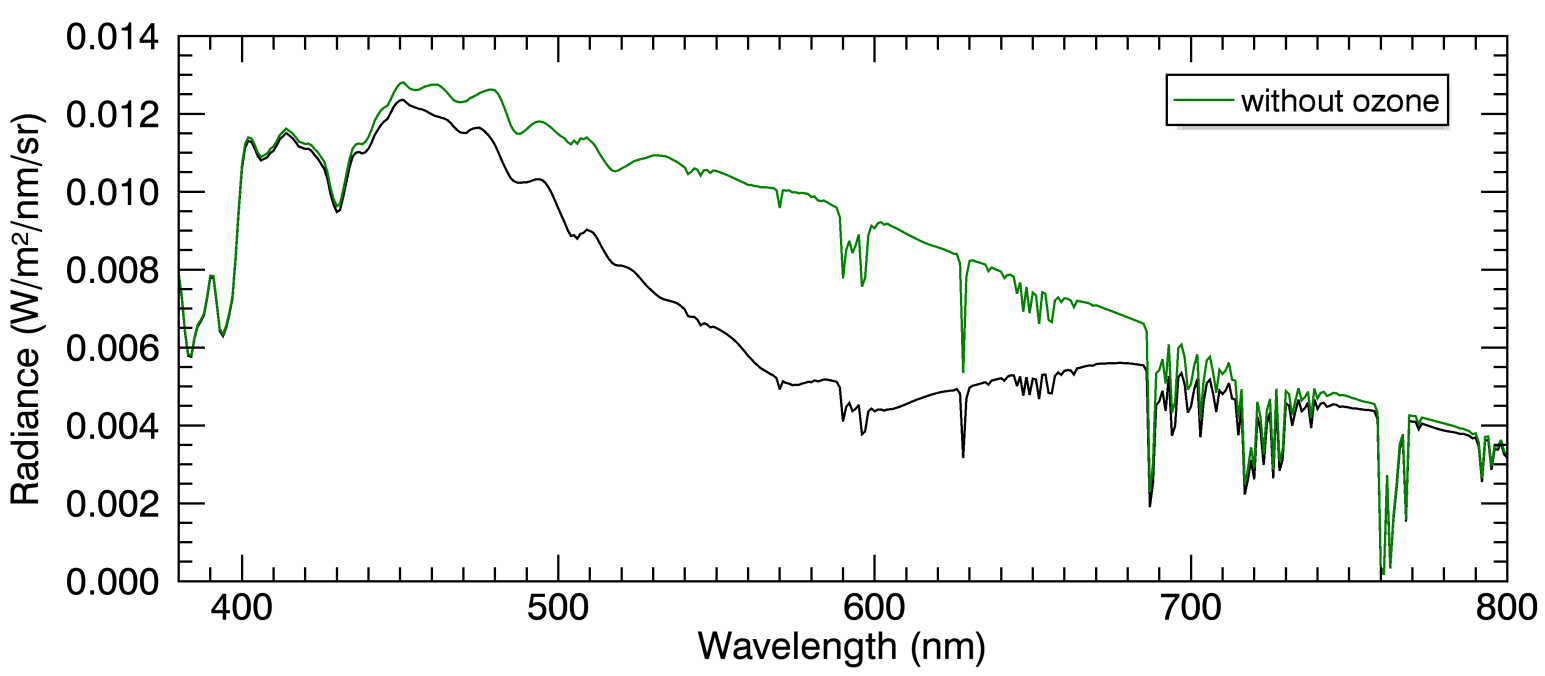

**Figure 4.** Scattered solar radiation spectra for SZA = 90°, SAA = 0° and VZA = 0° with TOC = 300 DU (black line) and TOC = 0 DU (green line).

Summarising, the results are in good agreement with the work of Hulburt (1953) and they significantly extend Hulburt's work in different ways. Ozone has been shown to strongly affect the colour of the sky not only for the zenith viewing geometry and SZA = 90°, but also for larger VZAs. In addition, it was shown that ozone also plays a role for the sky colour for smaller values of the SZA with the ozone influence increasing with both increasing VZA and SZA.

### 3.1 Influence of the total ozone column

For all calculations shown so far, a TOC of 300 DU was assumed. Figure 5 shows the resulting relative differences (compare equation 6) for SZAs = 10° (first row), 50° (second row) and 90° (third row) with TOC = 500 DU (left panels) and TOC = 100 DU (right panels). TOC values of 100 DU and 500 DU were chosen, because this range essentially covers all possible values occurring in the Earth's atmosphere.

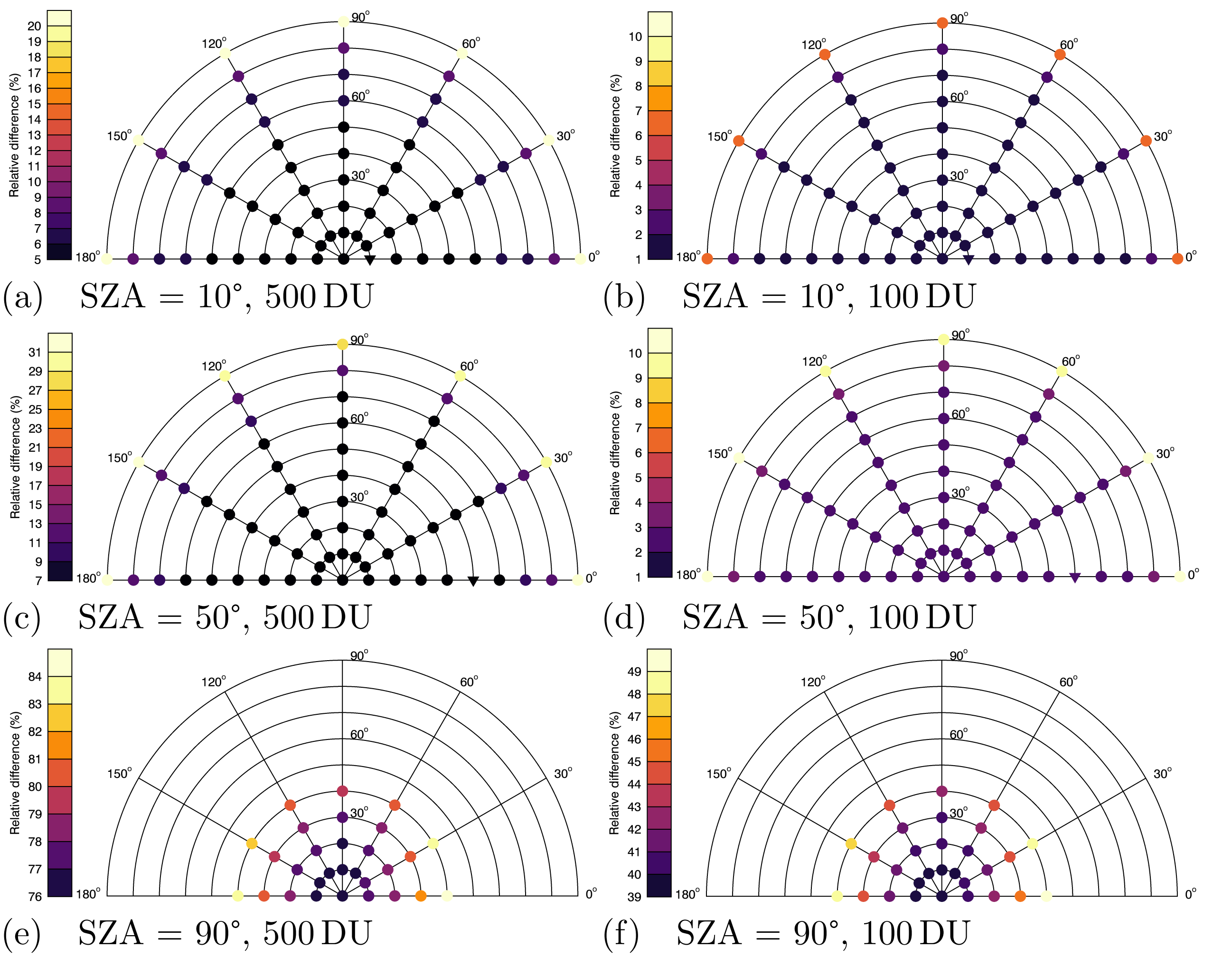

**Figure 5.** First row: Polar diagrams for SZA = 10°, TOC = 500 DU (a) and TOC = 100 DU (b). Second row: Polar diagrams for SZA = 50°, TOC = 500 DU (c) and TOC = 100 DU (d). Third row: Polar diagrams for SZA = 90°, TOC = 500 DU (e) and TOC = 100 DU (f) with the relative differences colour-coded.

With a TOC of 100 DU and SZA = 10°, the contribution of ozone is reduced to 1 % at the zenith (up to VZA = 70°) and 6 % at the horizon (panel (b) of Fig. 5). Assuming a larger TOC (panel (a) of Fig. 5), the ozone contribution increases to 5 % at the zenith (up to VZA = 50°) and 20 % at the horizon. In comparison, the ozone contribution for TOC = 300 DU is 3 % at the zenith and 15 % at the horizon (compare panel (a) of Fig. 3). At a SZA of 50° (second row of Fig. 5), the influence of ozone increases as expected due to the longer light path through the atmosphere. For TOC = 500 DU (c), the relative differences range from 7 % (VZA = 0°) to ≈ 31 % (VZA = 90°). For a smaller TOC (100 DU) (d), the relative differences are 2 % for VZA = 0° and ≈ 10 % for VZA = 90°. The exact values of the relative difference at VZA = 90° depend on the SAA.

While the contribution of ozone to the blue colour of the zenith is 66 % for SZA = 90° and 300 DU (as illustrated in panel (f) of Fig. 3), the contribution increases to 76 % for 500 DU (e) and correspondingly decreases to 39 % for 100 DU (f) (see the third row of Fig. 5). The relative differences for the VZAs 10° - 90° vary with the SAA. For 100 DU (f), the maximum value is 49 % and for 500 DU (e) 84 % for SAA = 0° and VZA = 40° – note that the approach does not work for larger VZAs.

The plots show that the contribution of ozone to the blue colour of the sky for different viewing geometries depends strongly on the assumed TOC. The contribution of ozone at 100 DU is nearly negligible for small SZAs (see panel (b) and (d) of Fig. 5). As expected, the values of the relative difference, i.e. the influence of ozone, increase for larger TOCs (e.g. 500 DU) and decrease for smaller TOCs (e.g. 100 DU) – compare Fig. 6.

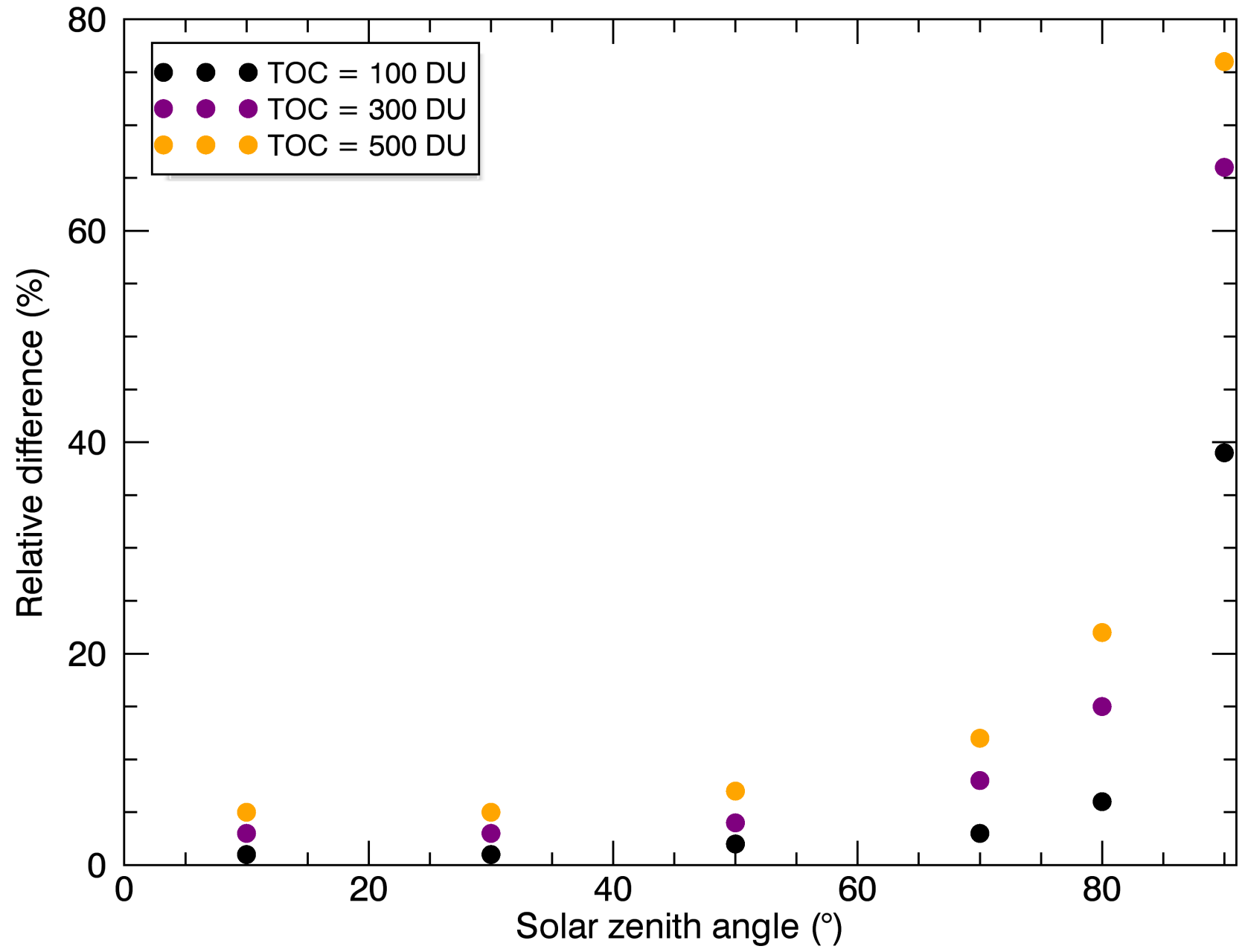

**Figure 6.** Relative difference (see equation 6) as a function of the SZA for different TOCs: 100 DU, 300 DU and 500 DU. The viewing geometry is the following: VZA = 0°, SZAs = 10°, 30°, 50°, 70°, 80°, 90° and SAA = 0°.

### 3.2 Aerosol effects

In order to test how the contribution of ozone to the blue colour of the sky changes with increased aerosol content, the following stratospheric aerosol scenario was assumed for the calculations: $r_m$ = 250 nm, S = 1.6 and a stratospheric aerosol optical depth (SAOD) of 0.1 at 550 nm. For comparison, the maximum globally averaged SAOD at 550 nm after the eruption of Mt. Pinatubo in June 1991 was about 0.15 (e.g., McCormick et al., 1995). The parameters of the tropospheric aerosols remain unchanged. The total AOD of both layers is 0.14 (at 550 nm) and a TOC of 300 DU was assumed. The symbols in the following polar diagrams shown in grey illustrate comparatively high values of the relative difference and are indicated separately for the sake of readability and clarity.

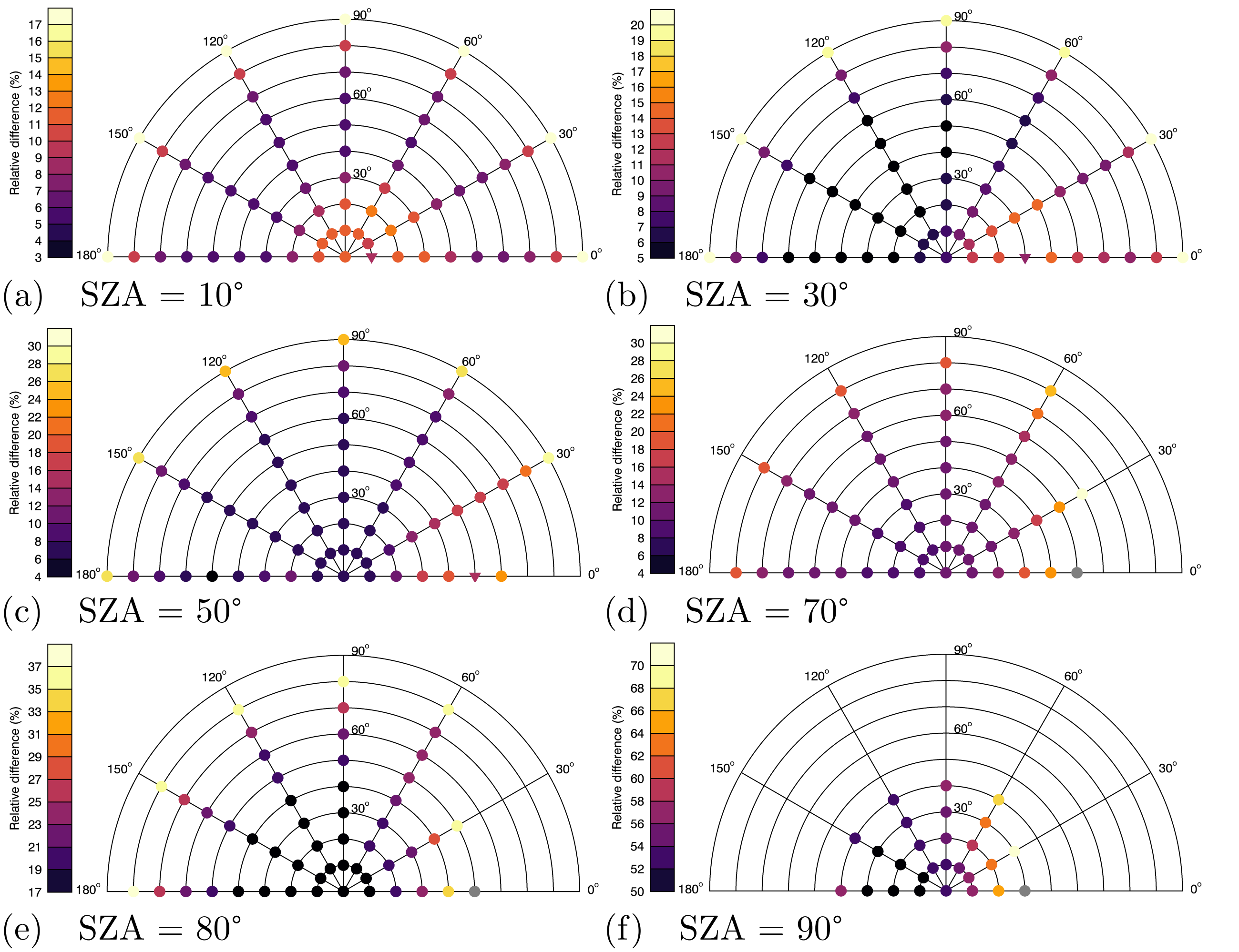

**Figure 7.** Polar diagrams for SZAs = 10° (a), 30° (b), 50° (c), 70° (d), 80° (e) and 90° (f) with the relative differences colour-coded. The calculations were carried out for the enhanced stratospheric aerosol content scenario with SAOD = 0.1 (550 nm). For SZA = 70° (d), SAA = 0° and VZA = 50° the value of the relative difference is 37 %. For SZA = 80° (e), SAA = 0° and VZA = 50° the value of the relative difference is 52 %. For SZA = 90° (f), SAA = 0° and VZA = 30° the value is 76 %.

The panels (a) and (b) of Fig. 7 show the results for SZA = 10° (a) and SZA = 30° (b). For both SZAs, the relative differences (compare equation 6) depend strongly on the SAA. While no SAA-dependent change in ozone contribution is observed in Fig. 3 for SZAs = 10° (a) and 30° (b) ("baseline aerosol scenario"), here the relative differences can vary with the SAA by up to 3 % units. For SZA = 10°, ozone contributes 11 % to the blue colour at the zenith. For SZA = 30° the contribution decreases to 8 %, but the maximum contribution (considering all VZA-SAA combinations) of ozone is still larger than for the SZA of 10°.

At the horizon, the relative differences are 17 % for a SZA of 10° and 20 % for SZA = 30° and SAAs of 0°, 30°, 150° and 180°.

The relative differences for a SZA of 50° (c) range from 6 % at VZA = 0° to ≈ 28 % (SAA = 30°) – 25 % (SAA = 90°) at VZA = 90°. In comparison, the ozone contribution at the zenith for a SZA of 70° (d) is 10 %. The grey symbol at SZA = 70°, SAA = 0° and VZA = 50° corresponds to a relative difference of 37 %. With 37 % the ozone contribution to the blue colour of the sky is comparatively large for this viewing geometry. For a SAA of 0°, the ground-based observer looks in the sun-ward direction, but with VZA = 50° not directly into the Sun. A final explanation for this high value cannot be given at this point.

The panels (e) and (f) of Fig. 7 show results for the calculations with the Sun 10° above the horizon (e) and at the horizon (f). The ozone contribution for SZA = 80° and VZA = 0° is 17 % here and increases to 36 % (SAAs 0° – 150°) at VZA = 80°. The viewing geometry of SZA = 80°, SAA = 0° and VZA = 50° corresponds to a value of 52 %. For SZA = 90° a significant increase in ozone influence is also observed for the enhanced stratospheric aerosol scenario. Here the contribution of ozone to the blue colour at the zenith is 53 % and at VZA = 40° 67 % (maximum value over all SAAs).

The calculations for SZA = 90° with the "baseline aerosol scenario" (panel (f) of Fig. 3) and the enhanced stratospheric aerosol content scenario (panel (f) of Fig. 7) show that more aerosols reduce the influence of ozone on the blue colour of the sky at and near the zenith. For instance, the value of the relative difference for the "baseline aerosol scenario" at the zenith is 66 %, which is reduced to 53 % with more aerosols and the specific aerosol particle size distribution assumed here. However, this is only the case for the SZA of 90°, since at the other SZAs shown here, the contribution of ozone is larger with more aerosols (compare, e.g., panel (a) of Fig. 3 and panel (a) of Fig. 7). Nevertheless, the differences between the relative differences of both scenarios decrease with increasing VZA (exact values also depend on the SAA) and SZA – especially for the SZAs of 10° – 80°. This supports the conclusion, that the light path through the atmosphere, which increases with increasing SZA, is crucial for this effect. Therefore, the influence of an increased aerosol content on the ozone contribution depends also on the position of the Sun and the exact viewing geometry.

Simulations were also carried out for two other aerosol scenarios, one with $r_m$ = 450 nm, S = 1.6, SAOD = 0.1 (550 nm), and the other with $r_m$ = 250 nm, S = 1.6, SAOD = 0.2 (550 nm), which lead to similar results for SZA = 90° (not shown).

In Adams et al. (1974) they concluded that with "ten times normal aerosol amount" (vertical optical thickness between 2 and 3.5) the blue of the zenith and near the zenith sky decreases, i.e. the spectral purity is reduced. The spectral purity indicates how monochromatic a colour is, e.g. a point near the "white point" in the CIE chromaticity diagram has a low spectral purity, whereas a point near the spectral arc of the CIE chromaticity diagram has a high spectral purity. Adams et al. performed these calculations only for the SZAs of 90° – 96° and considered just single scattering, but the shift to larger $x$ values can also be observed in the present calculations for the enhanced aerosol content scenario at all SZAs (not shown).

For the "baseline aerosol scenario", which corresponds to stratospheric aerosol background conditions, it is a valid approximation that the remaining contribution, besides the calculated ozone contribution, is due to Rayleigh scattering. This cannot be directly concluded for the enhanced stratospheric aerosol content scenario, since aerosols also have a contribution, as shown above.

In addition, we also tested the effect of the surface albedo and the height of the observer. A test with different values of the surface albedo (0.1 – 0.5) led to a maximum relative difference of the $x$, $y$ chromaticity coordinates of less than 1 %. For an observer at an altitude of 10 km, the contribution of ozone to the blue colour of the sky is 3 % for SZA = 10°, VZA = 0° and for VZA = 90° 16 %. For SZA = 90° and VZA = 0° the contribution is 78 %. With an observation height of 10 km an increase of the ozone contribution to the blue colour of the sky can be seen (compare with Fig. 3), which is consistent with expectations, since at an altitude of 10 km most of the ozone is still above this altitude and Rayleigh scattering is reduced.

## 4 Conclusions

With the radiative transfer model SCIATRAN, the CIE XYZ 1931 colour system and the new approach used in the present work we quantified the contribution of ozone to the blue colour of the sky for different viewing geometries. We were able to demonstrate quantitatively that the blue colour of the sky cannot be solely attributed to Rayleigh scattering. The calculations show that ozone contributes to the blue colour of the sky also beyond the zenith viewing geometry and SZA = 90°. The calculations show that the exact contribution of ozone is highly dependent on the assumed TOC. Moreover, the influence of ozone increases with increasing TOC, VZA and SZA. Ozone also contributes to the blue sky at small SZAs, although the contribution is found to be minor. In addition, the results for SZA = 90° are in good agreement with Hulburt's estimation of 2/3 ozone influence at the zenith at sunset. With more aerosols, the ozone contribution at the zenith is reduced to 53 % for SZA = 90°. Overall, the study of the influence of enhanced aerosol shows complex behaviour, as the ozone contribution is larger at small SZAs than in the "baseline aerosol scenario" and smaller at SZA = 90°. Calculations with different values of the surface albedo lead to minor effects on the ozone contribution to the blue colour of the sky. Furthermore, at an observation height of 10 km, an increase of the ozone influence is observed.

*Code and data availability.* The radiative transfer model SCIATRAN can be downloaded from the following link: https://www.iup.uni-bremen.de/sciatran/. The solar irradiance spectrum data are available at LASP (2003).

*Author contributions.* CvS outlined the project and AL carried out the SCIATRAN simulations with guidance by AR. AL wrote an initial version of the paper. All authors discussed, edited and proofread the paper.

*Competing interests.* The authors declare that they have no competing interests.

*Acknowledgements.* We are indebted to the Institute of Environmental Physics of the University of Bremen – particularly to Vladimir Rozanov and John P. Burrows FRS – for access to the SCIATRAN radiative transfer model. This study was enabled by the collaborations within the DFG research unit Volimpact (FOR 2820, grant no. 398006378).

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
