# Peer review of "Revisiting the question "Why is the sky blue?""

_EGUsphere, 2023_

## Author Comment (AC1)

**Replies to comments by reviewer 1**

**Comment:** The manuscript by Lange et al. focuses on the question "Why is the sky blue?". Using radiative transfer modelling, previous findings published in 1953 are confirmed and the impact of Rayleigh scattering and ozone absorption on the colour of the sky is investigated quantitatively.
The manuscript elucidates the reasons for the common phenomenon of a blue sky and confirms that ozone absorption has a significant contribution. Therefore, I feel that the subject of the manuscript is suitable for publication in ACP, although this work probably does not represent not cutting-edge science.

**Reply:** We thank the reviewer for his/her constructive and helpful comments. We tried to answer every comment in an appropriate way.

**Comment:** The metric used in this study, namely the distance of the chromaticity coordinates to the white point, does not seem to be suitable for a quantification of the "blueness" of the sky. It only tells whether the sky is less white, but it could as well be more green or red if the distance to the white point increases. Instead, I suggest to simply use z value to quantify blueness. Furthermore, the usage of relative differences (Equation 6) leads to very large values if the sky colour is close to the white point even if the absolute change is small. I would therefore suggest to use absolute differences, in particular for the polar diagrams shown in Figures 3, 5 and 7. Using absolute values would also avoid that many data points have to be dismissed, as it is currently the case for data near the white point.

**Reply:** Thank you for this detailed comment and the suggestions.
As explained in the manuscript (e.g., L24 – L25), the goal is to quantitatively determine the contribution of ozone to the blue colour of the sky. So the cases where the sky is no longer blue are not analysed. The check is done manually.
1. To the point about the z value: Each colour within the CIE chromaticity diagram has a z value, which means that the information of the z value is not unambiguous. For example: SZA = 90°, VZA = 0°, TOC = 300 DU: z = 0.474 (resulting colour: blue); SZA = 90°, VZA = 90°, SAA = 0°, TOC = 300 DU: z = 0.166 (resulting colour: orange). This means that criteria for applicability and thus for characterising the "blueness" would have to be considered here as well. Furthermore, the z value by itself is an "abstract" quantity that is not essential for the colour representation in the two-dimensional CIE chromaticity diagram, since this is based exactly on the application of the x,y chromaticity coordinates. Each colour in the CIE chromaticity diagram is characterised by the x,y chromaticity coordinates.
2. To the point about the absolute differences: The paper is exactly about the relative difference, i.e. the contribution of ozone to the blue colour of the sky (in %). This is the main point of the paper and we therefore want to keep showing relative differences in the polar diagrams. Nevertheless, we think that providing absolute differences is a good idea and we have now added the absolute difference to the essential points of the paper:
- **L146 – L148:**
It now says: "For SZA = 10° (panel (a) of Fig. 4) small relative differences are found as expected,

i.e. a minor influence of ozone on the blue colour of the sky, which increases with increasing VZA from 3 % (VZA = 0°) to 15 % (VZA = 90°), with the absolute differences of 0.004 (VZA = 0°) and 0.0065 (VZA = 90°). ”

- **L149 − L150:**

It now says: ”Similar results are observed for SZA = 30° (panel (b) of Fig. 4) with 3 % relative difference up to VZA = 50° and 17 % at VZA = 90° (absolute differences: 0.004 (VZA = 0°) and 0.007 (VZA = 90°)).”

- **L160 − L163:**

It now says: ”For SZA = 90° the relative differences and thus the contribution of ozone increases significantly. At the zenith, ozone contributes 66 % to the blue colour (absolute difference: 0.065).”

- **L179 − L180:**

It now says: ”With a TOC of 100 DU and SZA = 10°, the contribution of ozone is reduced to 1 % at the zenith (up to VZA = 70°) and 6 % at the horizon (panel (b) of Fig. 6). The absolute differences are 0.001 for VZA = 0° and 0.0025 for VZA = 90°.”

- **L180 − L182:**

It now says: ”Assuming a larger TOC (panel (a) of Fig. 6), the ozone contribution increases to 5 % at the zenith (up to VZA = 50°) and 20 % at the horizon (absolute differences: 0.006 (VZA = 0°) and 0.0095 (VZA = 90°)).”

- **L188 − L190:**

It now says: ”While the contribution of ozone to the blue colour of the zenith is 66 % for SZA = 90° and 300 DU (as illustrated in panel (f) of Fig. 4), the contribution increases to 76 % for 500 DU (e) (absolute difference: 0.108) and correspondingly decreases to 39 % for 100 DU (f) (absolute difference: 0.022) (see the third row of Fig. 6).”

- **L218 − L220:**

It now says: ”Here the contribution of ozone to the blue colour at the zenith is 53 % (absolute difference: 0.062) and at VZA = 40° 67 % (maximum value over all RAAs). ”

**Comment:** Each of the panels in Fig. 3, 5 and 7 have different colour scales, which makes a quantitative comparison very difficult for the reader. All panels should have the same colour scale.

**Reply:** As illustrated in the plot below (left panel: panel (d) from Figure 3 (now Figure 4), right panel: same plot with adjusted colour scale (covering a range from 2 % – 75 %)), with the wide range of values of the relative difference it is not possible to use the same colour scale for all figures, as this makes it almost impossible to interpret the figures. We appreciate the reviewers point, but in this case this would not work.

[Figure]

**Comment:** Why has the influence of ozone on the colour of the sky not been investigated for SZA > 90°, as in previous studies?

**Reply:** We focused on SZA ≤ 90°, because there are already several studies on the ozone influence for SZAs > 90°, i.e., twilight conditions as indicated in the manuscript (L45 – L51) and for SZA < 90° essentially no studies are available.

**Comment:** Specific comments
L10: The statement that the contribution of ozone increases with increasing VZA is already explained in L6 and L7 and can therefore be removed.

**Reply:** Thank you for the comment. L6 and L7 say: "For small solar zenith angles, the influence of ozone on the blue colour of the sky is minor, as expected. However, the effect of ozone increases with increasing solar zenith angle." L10 says: "The calculations also show that the contribution of ozone increases with increasing viewing zenith angle and total ozone column." Probably there is a misunderstanding here, because in L6 and L7 we are talking about the solar zenith angle (SZA) and in L10 about the viewing zenith angle (VZA).

**Comment:** L79: Please specify the wavelength grid used for the simulation of the spectra.

**Reply:** This is a good point. L79 now says: "Output of the simulations performed here are the intensities of the radiation in a wavelength range of 380 - 800 nm with a wavelength grid of 1 nm".

**Comment:** Section 2.2: Since not every reader is familiar with the definition of the chromaticity coordinates, I suggest to add a graphical representation of the CIE colour matching functions. For example, Fig. 4 could be moved to section 2.2, and the three colour matching functions could be added to this graph. This would also make it clearer why the sky appears bluer in the presence of ozone, although its absorption maximum is in the green.

**Reply:** Thank you, we added an illustration of the CIE colour matching functions to Section 2.2. In addition, we modified L87 – L90: "Using the CIE colour matching functions $\overline{x}(\lambda)$, $\overline{y}(\lambda)$ and $\overline{z}(\lambda)$ (compare Fig. 2), which describe the spectral sensitivity of the cone cells of the human eye [...]".

**Comment:** Section 2.3: As already mentioned in the general comments, I feel that the z coordinate is a better proxy for the blueness of the sky than the distance to the white point, and absolute values would be more appropriate than relative ones. It is not defined how the distance is actually calculated. Is this the Euclidean distance?

**Reply:** Thank you, yes it is the Euclidean distance. We added this fact to L107 of the manuscript. As described in more detail above, the z value is not an explicit quantity, i.e. criteria for characterising the "blueness" of the sky would also have to be considered in this case. Furthermore, the use of the relative difference fulfils exactly the goal of the paper, i.e. the question of the quantitative contribution of ozone to the blue colour of the sky. Nevertheless, we have included information about the absolute difference in the main parts of the paper (also described in more detail above).

**Comment:** It is stated that the method described here is not applicable in all cases. What exactly are the criteria for the applicability of the method and for dismissing particular data points?

**Reply:** The method described in the paper is not applicable if the resulting colour is not blue. As described in L113 – L116 of our manuscript: "The method is not applicable for all cases. The right panel of Fig. 2 (now Fig. 3 due to the insertion of the illustration of the CIE colour matching functions to Section 2.2 – compare with the comment and reply above)) shows for e.g. VZAs = $60°$ and $70°$ viewing geometries where the method is inapplicable. That is generally the case for chromaticity coordinates corresponding to the green, yellow, orange and red colour regions of the CIE chromaticity diagram and can apply to one data point or both (with and without ozone) – this covers all cases occurring in this work but not every theoretically possible one." And L118 – L120: "Note that the fact that the method is not applicable in all cases does not affect the goals of the study, i.e. to determine the contribution of ozone to the blue colour of the sky, because the sky is not blue anymore in the cases that cannot be studied." This means, that in those cases that are not analysed, the resulting colour, i.e the colour of the sky, is no longer blue. The selection and verification was done manually.

**Comment:** L129: I do not understand the purpose of normalising the aerosol number density to 1 particle/cm$^3$.

**Reply:** We think there is a misunderstanding here, because we did not perform the normalisation of the aerosol number density to 1 particle/cm$^3$, but the aerosol number density is so available in Shettle and Fenn (1979). To avoid this misunderstanding, we have reworded this sentence.

**Comment:** L203: It would be good if you could add a discussion on the reasons for the increase of sensitivity to ozone with higher stratospheric aerosol load. Is this due to an increase in average scattering altitude, yielding a longer stratospheric light path, or due to a reduced fraction of Rayleigh scattering in the presence of stratospheric particles?

**Reply:** Thank you for the comment. The increase of the ozone influence observable with the "enhanced stratospheric aerosol content scenario" is only valid for the SZAs 10° – 80°. For SZA = 90° a reduction of the ozone contribution on the blue colour of the sky at and near the zenith is observed (L217 – L225 of our manuscript). We cannot give any final explanations for this at this point, but we added the following possible explanations to the manuscript: " Note, that there is no final explanation for this effect at this point, but the following may represent a possible explanation. With SZAs of 10° – 80° and the enhanced aerosol content of the stratospheric aerosol layer (10 – 25 km), the scattered sunlight perceived by the observer on the Earth's surface comes mainly from this altitude and thus has a longer light path through the stratospheric ozone layer, resulting in higher relative differences, i.e. higher ozone contributions. With the Sun at the horizon, the scattered sunlight comes mainly from higher altitudes, resulting in a reduction of the ozone contribution. This is a potential explanation that we cannot prove conclusively.".

**Comment:** Conclusions: In order to give proper credit to previous work, it would be good if you could make clear in the conclusions that this work is a confirmation of previous studies on the influence of ozone on the colour of the sky, although your study provides a better quantitative assessment.

**Reply:** We added the following to the conclusions: "Therefore, our work represents an additional confirmation of previous studies on the influence of ozone on the blue colour of the sky, although our work is based on a new quantitative method." (L248 – L250). However, our study is not only an additional confirmation of existing studies, it also goes deeper into the content (e.g. more viewing geometries) – as mentioned in the conclusions.

**Comment:** Technical comments
L63: I suggest to replace the term SAA with RAA (relative azimuth angle), since it represents the azimuth angle between the viewing direction and the direction of the Sun, whereas SAA usually describes the position of the Sun relative to the North.

**Reply:** This is a good point, we changed that.

**Comment:** Equation 6: the term "relative difference" is inappropriate for a variable name. According to existing conventions (Cohen et al, 2008), variable names should preferably consist of a single letter, such as "r". Multiplying with the term "100 %" on the left side is redundant since $100/100 = 1$.
References
E.R. Cohen, T. Cvitas, J.G. Frey, B. Holmström, K. Kuchitsu, R. Marquardt, I. Mills, F. Pavese, M. Quack, J. Stohner, H.L. Strauss, M. Takami, and A.J. Thor, "Quantities, Units and Symbols in Physical Chemistry", IUPAC Green Book, 3rd Edition, 2nd Printing, IUPAC & RSC Publishing, Cambridge (2008)

**Reply:** We corrected equation 6 and the corresponding sentence above. Furthermore, we adjusted the labelling of the colour scales of the polar diagrams as follows: "relative difference r (%)" (compare with the polar diagrams above).

**Comment:** L139: The sentence starting with "The radius of the polar diagram corresponds to..." should be moved to the caption of Figure 3.

**Reply:** We added this sentence to the caption of Figure 3 (now Figure 4). The caption of Figure 4 now says: "Polar diagrams for SZAs = 10° (a), 30° (b), 50° (c), 70° (d), 80° (e) and 90° (f) with the relative differences colour-coded. The radius of the polar diagram corresponds to the VZA and the angle to the RAA (marked at the outer edge of the diagram). The missing values are due to the non-applicability of the method as described above."

**Comment:** Figures 3, 5 and 7 should all have the same colour scale in all panels (see general comments).

**Reply:** As explained in more detail above, with the wide range of the values of the relative difference, it is impossible to use the same colour scale for all figures and still ensure readability and interpretability.

**Comment:** L151: "light path" –> "light paths"

**Reply:** Thank you, corrected.

**Comment:** L155: "a SZA" –> "an SZA"

**Reply:** Thank you, changed.

**Comment:** L160: "increase" –> "increases"

**Reply:** Thank you, also changed.

**Comment:** L162 and caption of Fig. 6: It doesn't make sense to specify the SAA for zenith measurements (VZA = 0°)

**Reply:** This is a good point, we corrected that.

**Comment:** L204: delete "units"

**Reply:** Thank you, changed.

**Comment:** L204: "aerosols" –> "stratospheric aerosols"

**Reply:** Thank you for the comment. Unfortunately, we could not find "aerosols" in L204 or in the lines above/below. But we added "stratospheric" to the following lines: L8, L222, L224, L226, L234

and L235.

**Pre-Production Review**

Regarding the figure 1: please ensure that the colour schemes used in your maps and charts allow readers with colour vision deficiencies to correctly interpret your findings. Please check your figures using the Coblis – Color Blindness Simulator (https://www.color-blindness.com/coblis-color-blindness-simulator/) and revise the colour schemes accordingly.

**Reply:** We revised the colour scheme of Figure 1 as follows.

[Figure]

---

## Author Comment (AC2)

**Replies to comments by reviewer 2**

**Comment:** The paper presents a radiative transfer modelling study to explain the blueness of the sky. In most cases the color is due to Rayleigh scattering, but during twilight also ozone absorption plays a role. In 1953 E. Hulbert claimed based on a simplified modeling approach that the color during sunset is to 1/3 caused by Rayleigh scattering and to 2/3 caused by ozone absorption. In this work the color of the sky is investigated quantitatively by simulating spectra under various conditions and converting those to the CIE color space. The study basically confirms the result by E. Hulbert. The paper is clearly written, well understandable with appropriate number of figures. Altough the result is not really new because it just confirms what is expected, it provides some new insights. Therefore I recommend to publish the paper after minor revisions.

**Reply:** We thank the reviewer for his/her constructive and helpful comments. We tried to answer every comment in an appropriate way.

**Comment:** General comments:
1. Eq. 6, Quantification of color difference: Wouldn't it be better to use the absolute valud of the difference vector between the vectors in the CIE diagram instead of distances to the white point, e.g.

abs((x,y)ozone-(x,y)) ?

You explain that if a point is in another direction (e.g. red) you can not evaluate the result. If you take the difference as a vector, couldn't you also evaluate the reddish points during sunset?

**Reply:** Since the paper deals with the ozone influence on the blue colour of the sky, there is no need to evaluate data points where the resulting colour is no longer blue. As already answered in more detail in the first report, it is exactly about the relative difference, i.e. how large is the contribution of ozone to the blue colour of the sky (in %). Nevertheless, we have now added the absolute differences to the main points of the paper.

**Comment:** 2. Impact of polarization (l. 71): 1% seems relatively small. Is this result in line with Mishchenko et al 1994? Probably the effect would be largest for a scattering angle around 90° and for large AOD, e.g. SZA=90° and VZA=0°. Has this been tested?
Reference:
Mishchenko, M.I., A.A. Lacis, and L.D. Travis, 1994: Errors induced by the neglect of polarization in radiance calculations for Rayleigh-scattering atmospheres. J. Quant. Spectrosc. Radiat. Transfer, 51, 491-510, doi:10.1016/0022-4073(94)90149-X.

**Reply:** Please note that in the manuscript we talk about the influence on the x,y chromaticity coordinates and not about the radiance spectra themselves. Simulations considering polarisation for SZA = 90° and different viewing geometries led to similar results with, e.g., VZA = 0° of slightly more than $1\%$ relative difference of the $x$, $y$ chromaticity coordinates. In agreement with Mishchenko et al. (1994), our calculations also show a larger maximum relative difference in the intensities at SZA

= 90° and VZA = 0° (here: ≈ 7%). In our manuscript we speak only of the relative difference of the $x$, $y$ chromaticity coordinates, since these form the basis of the method and are thus crucial for the presented results.

**Comment:** 3. l. 124: AOD=0.04 is quite small. Is it a typical value for Greifswald or rather at the low end?

**Reply:** The tropospheric aerosol optical depth of 0.04 at 550 nm is in the range of AOD values observed with the AERONET photometer at the Institute of Physics of the University of Greifswald (as mentioned in L126 – L128). Smaller values of 0.03 (at 550 nm), for example, have also been observed. However, the tropospheric aerosol optical depth is highly variable (as mentioned in L132 – L133).

**Comment:** 4. l. 210ff.: "With 37% the ozone contribution to the blue colour of the sky is comparatively large for this viewing geometry. For a SAA of 0°, the ground-based observer looks in the sun-ward direction, but with VZA = 50° not directly into the Sun. A final explanation for this high value cannot be given at this point."
Have you looked at the scattering phase function of the aerosol particles? This could explain why you get the largest contribution for this particular geometry.

**Reply:** This is a very plausible idea and we also believe that the phase function plays an important role for this geometry (although we cannot strictly show it). We added a sentence to the manuscript discussing this possibility.

**Comment:** 5. A general RT modelling question: How is refraction modelled in combination with polarization. For scalar RT the ray is bended according to Snell's law, is this valid when polarization is considered? Or do you need to use the Fresnel equation when the ray crosses a layer boundary? Could you provide a reference describing how this is treated?

**Reply:** Thank you for the comment. We do not have any publication to this topic, but refraction is treated in exactly the same way with the polarisation as without. This is just a geometrical ray bending according to Snell's law.